# Effects of Hyperbaric Oxygen Intervention on the Degenerated Intervertebral Disc: From Molecular Mechanisms to Animal Models

**DOI:** 10.3390/cells12162111

**Published:** 2023-08-21

**Authors:** Song-Shu Lin, Steve W. N. Ueng, Kowit-Yu Chong, Yi-Sheng Chan, Tsung-Ting Tsai, Li-Jen Yuan, Shih-Jung Liu, Chuen-Yung Yang, Hui-Yi Hsiao, Yi-Jen Hsueh, Chung-An Chen, Chi-Chien Niu

**Affiliations:** 1Department of Orthopaedic Surgery, Chang Gung Memorial Hospital, Taoyuan 333, Taiwan; lss1192001@yahoo.com.tw (S.-S.L.); chan512@cgmh.org.tw (Y.-S.C.); tsai1129@gmail.com (T.-T.T.); ycyfl@yahoo.com.tw (C.-Y.Y.); dwat5899@hotmail.com (C.-A.C.); 2Department of Nursing, Chang Gung University of Science and Technology, Taoyuan 33302, Taiwan; 3Hyperbaric Oxygen Medical Research Laboratory, Bone and Joint Research Center, Chang Gung Memorial Hospital, Taoyuan 33305, Taiwan; kchong@mail.cgu.edu.tw; 4College of Medicine, Chang Gung University, Taoyuan 333, Taiwan; 5Department of Medical Biotechnology and Laboratory Science, College of Medicine, Chang Gung University, Taoyuan 333, Taiwan; 6Department of Orthopaedic Surgery, E-Da Hospital, I-Shou University, Kaohsiung 82445, Taiwan; 7Department of Mechanical Engineering, Chang Gung University, Taoyuan 333, Taiwan; shihjung@mail.cgu.edu.tw; 8Center for Tissue Engineering, Chang Gung Memorial Hospital, Taoyuan 333, Taiwan; ivyhsiao@gmail.com (H.-Y.H.); t6612@seed.net.tw (Y.-J.H.); 9Department of Biomedical Science, College of Medicine, Chang Gung University, Taoyuan 333, Taiwan; 10Department of Ophthalmology, Chang Gung Memorial Hospital, Taoyuan 333, Taiwan

**Keywords:** microRNA, Wnt3a/β-catenin signaling, intervertebral disc, degeneration, hyperbaric oxygen

## Abstract

MicroRNA (miRNA) 107 expression is downregulated but Wnt3a protein and β-catenin are upregulated in degenerated intervertebral disc (IVD). We investigated mir-107/Wnt3a-β-catenin signaling in vitro and in vivo following hyperbaric oxygen (HBO) intervention. Our results showed 96 miRNAs were upregulated and 66 downregulated in degenerated nucleus pulposus cells (NPCs) following HBO treatment. The 3′ untranslated region (UTR) of the Wnt3a mRNA contained the “seed-matched-sequence” for miR-107. MiR-107 was upregulated and a marked suppression of Wnt3a was observed simultaneously in degenerated NPCs following HBO intervention. Knockdown of miR-107 upregulated Wnt3a expression in hyperoxic cells. HBO downregulated the protein expression of Wnt3a, phosphorylated LRP6, and cyclin D1. There was decreased TOP flash activity following HBO intervention, whereas the FOP flash activity was not affected. HBO decreased the nuclear translocation of β-catenin and decreased the secretion of MMP-3 and -9 in degenerated NPCs. Moreover, rabbit serum KS levels and the stained area for Wnt3a and β-catenin in repaired cartilage tended to be lower in the HBO group. We observed that HBO inhibits Wnt3a/β-catenin signaling-related pathways by upregulating miR-107 expression in degenerated NPCs. HBO may play a protective role against IVD degeneration and could be used as a future therapeutic treatment.

## 1. Introduction

Low back pain (LBP) is directly associated with reduced quality and expectancy life [1]. Intervertebral disc (IVD) degeneration is one of the main causes of LBP [2,3]. The pathogenesis of IVD degeneration (IDD) is a complex disease, including excessive mechanical loading, trauma, inflammation, genes, aging, and obesity [2,4,5,6,7,8]. The process of IDD is believed to have a biochemical basis, with enhanced matrix degradation and inhibition of nuclear proteoglycan synthesis caused by chemical mediators that may include nitric oxide (NO), interleukin (IL)-1, and matrix metalloproteinases (MMPs) [9,10]. A common characteristic of IDD is the imbalanced synthesis and catabolism of extracellular matrix (ECM) proteins. An elevated level of MMPs has been correlated with human progressive disc degeneration [11] and in animal models of IDD [12].

Wnt (wingless and int-1) signaling is activated upon binding of several members of the Wnt protein family to the Frizzled/low-density lipoprotein receptor-related protein 5 or 6 (Fz-LRP5/6) receptor complex. This causes β-catenin stabilization and translocation to the nucleus, where it binds to the lymphoid enhancer factor (LEF) and T-cell factor (TCF) transcription factors to activate Wnt target gene expression [2]. Wnt proteins are important IVD cell regulatory factors. β-Catenin has been reported to be upregulated in disc tissue samples from patients with disc degeneration [3]. Activation of Wnt signaling suppresses the proliferation of nucleus pulposus cells (NPCs), promotes cellular senescence, and induces the expression of matrix MMPs 3, 7, and 9, suggesting that Wnt signaling triggers the process of IVD degeneration [13,14,15]. In addition, the expression of Wnt3a, GSK-3β, cyclin D1, and β-catenin was notably augmented in parallel with IDD progression [16].

MicroRNAs are endogenous non-coding small RNAs which can mediate gene regulatory events via pairing with the 3′-untranslated region (3′-UTR) of their target mRNAs and adjust their expression. MicroRNAs regulate various cellular processes including apoptosis, proliferation, and differentiation [17]. MiR-640 aggravates IDD via nuclear factor κB (NF-κB) and Wnt signaling pathway [18]. Overexpression of miR-185 alleviates IDD through inactivation of the Wnt/β-catenin signaling pathway and downregulation of Galectin-3 [19]. MiR-107 inhibited osteosarcoma cell proliferation and migration through inhibition of *β*-catenin signaling [20]. MiR-103/107 targeted Wnt family member 3a (Wnt3a) in preadipocytes [21]. MiR-107 was reported to be downregulated in IDD [22]. However, the function of miR-107 on Wnt3a regulation in IVD is not clear.

IVD is the largest avascular tissue in human and it receives oxygen and glucose through the cartilage endplate [23]. The flow of nutrients and metabolites is reduced during aging and degeneration of the IVD [24]. The hypoxic state of the NPCs [25,26] is in turn enhanced by the loss of CEP permeability during IVD degeneration [27,28]. The degenerated disc tissue has lower oxygen tension than a healthy one [29,30]. Degenerated discs have a hypoxic and inflammatory microenvironment leading to upregulation of catabolic factors and further degeneration [31]. Hyperbaric oxygen (HBO) treatment increased tissue/microvascular pO_2_ [32]. HBO inhibits the HMGB-1/RAGE signaling pathway by regulating miR-107 expression in human degenerative NPCs [33]. Because rat NPCs have been shown to exhibit increased levels of β-catenin under the hypoxic condition [19], we first investigated the effects of miR-107 on Wnt3a/β-catenin signaling and MMP expression in human degenerated NPCs following HBO intervention in vitro.

The IVD comprises nucleus pulposus (NP) and annulus fibrosus (AF). The extracellular matrix (ECM) of the NP comprises glycosaminoglycans (GAGs), proteoglycans (PGs), and type II collagen. Progressive reduction in PG content in the NPCs that deteriorates with age can lead to IVD degeneration. Keratan sulfate (KS) is a major component of cartilage PGs. Because the degradation of cartilage PGs in degenerative IVD causes a dramatic rise in levels of serum KS, measurement of serum KS levels may provide useful information about PG degradation occurring in an injured or degenerative IVD [34,35]. Serum KS serves as a potential biomarker of loading of the IVD [35]. We evaluated whether these serum cartilage metabolites can be used as biomarkers for mechanical loading of rabbit discs following HBO treatment.

In the present study, our data showed that HBO increased miR-107 expression in human degenerated NPCs, as assessed by way of microarray analysis and confirmed by real-time PCR. Furthermore, we used bioinformatics to identify putative target sequences for miR-107 in human Wnt3a and confirmed these via luciferase reporter assays. Moreover, the authors investigated the effects of HBO on the expressions of miR-107/Wnt3a/LRP 6/β-catenin-mediated signaling in human degenerated NPCs. Finally, we examined the effects of HBO on Wnt3a/β-catenin expression and serum KS levels in a rabbit degenerated IVD model.

## 2. Materials and Methods

### 2.1. Patients

Fresh disc tissues were harvested from 28 degenerated lumbar IVD patients undergoing spinal surgery. IVD degeneration grade was determined by evaluation of MRI. The experimental protocol was approved by the Human Subjects Institutional Review Board and the Institutional Animal Care and Use Committee of the Chang Gung Memorial Hospital. All patients agreed to undergo surgical intervention and informed consent was obtained.

### 2.2. Nucleus Pulposus Cell Isolation and Culture

Human NPCs were separated from the nuclear tissue by sequential enzymatic digestion, first with 0.4% pronase (Sigma, St. Louis, MO, USA) for 1 h and subsequently with 0.025% collagenase P (Boehringer Mannheim, Germany) and 0.004% DNase II (Sigma) at 37 °C overnight. After digestion, the cells were washed extensively with DMEM/F-12, then seeded in 3 fresh flasks at a density of 5000 cells/cm^2^ and incubated in a humidified atmosphere of 5% CO_2_ and 95% air until the cells attained confluence.

### 2.3. Exposure to Intermittent HBO

The cells were plated at 3 × 10^5^ cells per 100 mm culture dish in 10 mL of DMEM/F-12 containing 5% FBS. Control cells were maintained in 5% CO_2_/95% air at 1 atmosphere throughout the experimental protocol. All hyperoxic cells were exposed to 100% O_2_ at 2.5 ATA (atmospheres absolute) in a hyperbaric chamber (Cherng Huei Corporation, Taiwan) for 90 min per 48 h. A series of 3 HBO treatments was used. At 24 h after each treatment, conditioned media (CM) were collected and centrifuged at 1200× *g* for 5 min to remove debris, then stored at −70 °C until analysis.

### 2.4. MicroRNA Profiling

Total RNA was extracted from cells using mir Vana miRNA isolation kit (Ambion, Austin, TX, USA). MiRNA expression profiling was accomplished using TaqMan Human MicroRNA Array A Cards containing 384 mature human microRNAs (Applied Biosystems, Waltham, MA, USA) and an ABI 7900 Real-Time PCR System according to the manufacturer’s protocol. MiRNA expression profiling was performed on ten samples from five patients (with or without HBO treatment). Briefly, 3 μL of total RNA from each sample were reverse-transcribed using the TaqMan miRNA Reverse Transcription Kit (Applied Biosystems) and the stem-loop Megaplex Primer Pool Sets. A total of 7.5 μL of reaction mixture was immediately incubated under the following conditions: 40 cycles at 16 °C for 2 min, 42 °C for 1 min, 50 °C for 1 s, and 85 °C for 5 min. Then, 2.5 μL of the resultant Megaplex RT products were mixed with 2.5 μL of Megaplex PreAmp Primers and 12.5 μL of TaqMan PreAmp Master Mix. A volume of 25 μL of the reaction mixture was incubated using the following program: 95 °C for 10 min, 55 °C for 2 min, and 72 °C for 2 min, followed by 12 cycles at 95 °C for 15 s, 60 °C for 4 min, and 99.9 °C for 10 min. The pre-amplified cDNA was diluted with 0.1× TE (10mM Tris, pH 8.0, 1 mM EDTA) to 100 μL and used for PCR. The relative miRNA expression levels were calculated by the 2^−ΔΔCt^ method as follows: ΔCt (test) = Ct (miRNA of interest, test) − Ct (internal reference, test), ΔCt (calibrator) = Ct (miRNA of interest, calibrator) − Ct (internal reference, calibrator), ΔΔCt = ΔCt (test) − ΔCt (calibrator). The inter-individual variability of the efficiency of our procedures was controlled by spiking of U6 snRNA.

### 2.5. Real-Time PCR

TaqMan miRNA Assays (Applied Biosystems) were used to detect the expression levels of the mature miR-107. For the reverse transcription (RT) reactions, 10 ng of total RNA were mixed with the RT primer. RT reactions were performed at 16 °C for 30 min, 42 °C for 30 min, and 85 °C for 5 min, then maintained at 4 °C. Following the RT reactions, 1.5 μL of cDNA was used for a polymerase chain reaction (PCR) using 2 μL of TaqMan primers. The PCR was conducted at 95 °C for 10 min followed by 40 cycles of 95 °C for 15 s and 60 °C for 60 s in an ABI 7900 Real-Time PCR System (Applied Biosystems). The quantitative PCR results were analyzed and expressed as the relative miRNA level using U6 snRNA for normalization purposes. The fold change in the miRNA expression in each sample relative to the average expression in the control was calculated based on the threshold cycle (CT) value using the 2^−ΔΔCt^ method.

### 2.6. MiRNA Target Prediction and Dual-Luciferase Reporter Assays

Target Scan 7.2 and miRnalyze online software (version 1) were used to analyze the putative target genes of miR-107. The 3′UTR of Wnt3a containing the miR-107 binding site was cloned into pmirGLO Dual-Luciferase miRNA Reporter Vectors (Promega, Madison, WI, USA). A mutated 3′ UTR of Wnt3a was introduced into the potential miR-107 binding site. The reporter vectors containing the wild-type or mutant Wnt3a 3′ UTR were transfected into NPCs using Lipofectamine 3000 (Invitrogen, Carlsbad, CA, USA). Briefly, cells were seeded in six-well tissue culture plates at 2 × 10^5^ cells/per well in 2.5 mL Opti-MEM (Invitrogen) at 12 h before transfection. On the day of transfection, the cells were exposed to the reporter vectors/Lipofectamine 3000 mixtures containing 2.5 μg of the luciferase reporter plasmid DNA mixture. At 8 h after transfection, the transfection medium was changed to (DMEM)/F-12 culture medium with 5% FBS and the cells were exposed to HBO. After 48 h, the transfected cells were washed with PBS and harvested using a passive lysis buffer (Promega). Cell lysates (20 μL) were evaluated for luciferase activity using a Dual-Luciferase Reporter Assay Kit (Promega).

### 2.7. Transfection of NPCs with Anti-miRNAs and Analysis following HBO Intervention

NPCs were seeded into six-well plates at a density of 2 × 10^5^ cells/well in culture medium without antibiotics. The next day (day 1), cells were transfected with anti-miR-107 (100 nM; Ambion, USA) using RNAiMAX (Invitrogen, USA) and cultured in an incubator at 37 °C with 5% CO_2_. After 8 h of transfection, the culture medium was changed to DMEM/F-12 with 10% FBS, and the cells were exposed to HBO intervention. On days 3 and 5, the cells were re-transfected once and exposed to HBO. At 12 h after the third HBO treatment, cellular RNA was isolated using an RNeasy Mini Kit (Qiagen, Valencia, CA, USA) and reverse-transcribed into cDNA with the ImProm-II Reverse Transcription System (Promega, USA). For real-time PCR detection of Wnt3a transcripts, cDNA was analyzed on an ABI PRISM 7900 Sequence Detection System using TaqMan PCR Master Mix (Applied Biosystems, USA). The cycle threshold (Ct) values were obtained, and the data were normalized to β-actin expression using the 2^−ΔΔCt^ method to calculate the relative mRNA levels of each target gene.

At 24 h after the third HBO intervention, cells were washed with PBS and extracted using M-PER Mammalian Protein Extraction Reagent (Thermo, Rockford, IL, USA). For immunoblotting experiments, the proteins were separated via SDS-PAGE and transferred onto nitrocellulose membranes using a protein transfer unit (Bio-Rad, Hercules, CA, USA). After blocking with 10% nonfat milk, the membranes were incubated overnight at 4 °C with a 1000-fold dilution of mouse antibodies against Wnt3a (Cell Signaling, MA, USA). After washing, the membranes were further incubated for 2 h with 10,000-fold goat anti-mouse IgG (Calbiochem, San Diego, CA, USA) conjugated to horseradish peroxidase. The membranes were then washed and rinsed with ECL Detection Reagents (Amersham Pharmacia Biotech, Amersham, UK). The band images were photographed using Hyperfilm (Amersham). The intensity of the staining for Wnt3a and β-actin was quantified using an image analysis system (Image-Pro Plus 5.0; Media Cybernetics, Silver Spring, MD, USA).

### 2.8. Protein Extraction and Western Blot Analysis of Wnt3a, Phosphor-LRP6, LRP6, and Cyclin D1

Cells were plated at a density of 3 × 10^5^ cells per 100 mm culture dish in 10 mL of DMEM/F-12 containing 5% FBS. At 24 h after 3 HBO interventions, the cells were washed with PBS and extracted using M-PER Protein Extraction Reagent (Thermo, USA). The protein content was quantitated using a protein assay kit (Pierce Biotechnology, Rockford, IL, USA), separated by 7.5% SDS-PAGE, and transferred onto membranes using a transfer unit (Bio-Rad, USA). After blocking, the membranes were incubated with 1000-fold diluted rabbit antibodies against Wnt3a, phosphor-LRP6 (Cell Signaling), LRP6 (Abcam, Cambridge, UK), cyclin D1/2 (Merck, Darmstadt, Germany), mouse antibodies against β-catenin (Millipore, Temecula, CA, USA), and β-actin (Millipore). After washing, the membranes were further incubated for 2 h with 10,000-fold goat anti-mouse IgG (Calbiochem, USA) or goat anti-rabbit IgG (Millipore) conjugated to horseradish peroxidase. The membranes were then washed and rinsed with ECL detection reagents (Millipore). The bands were photographed using ECL Hyperfilm (Amersham Pharmacia UK) and their intensity was quantified using an image-analysis system (Image-Pro Plus 5.0).

### 2.9. TOP/FOP Flash Luciferase Assay

The activity of the Wnt signaling pathway was detected by a Wnt signal reporter assay using the TOP/FOP TCF reporter kit (Millipore, USA). Cells were seeded in 24-well tissue culture plates at 5 × 10^4^ cells/well in 0.5 mL of Opti-MEM (Invitrogen) at 12 h before transfection. On the day of transfection (day 1), 900 ng of the TOP flash or FOP flash construct (Millipore) together with 100 ng of the pGL4.74[hRluc/TK] plasmid (Promega) was used to transfect the cells in each well. The pGL4.74[hRluc/TK] plasmid containing the Renilla luciferase gene was used as an internal control for normalizing the transfections. Transient transfections using Lipofectamine LTX and PLUS reagent (Invitrogen) were performed according to the manufacturer’s instructions. Eight hours after transfection, the transfection medium was changed to DMEM/F-12 containing 5% FBS, and the cells were exposed to HBO. On days 3 and 5, the cells were re-transfected once and exposed to HBO as described above. After an additional 24 h of culturing, the cells were washed with PBS and harvested using 100 μL/well of passive lysis buffer (Promega). The cell lysates (20 μL) were evaluated for luciferase activity using a Dual-Luciferase Reporter Assay Kit (Promega). Luciferase activity was measured according to the manufacturer’s instructions.

### 2.10. Preparation of Cytosolic and Nuclear Fractions for β-Catenin Detection

At 24 h after 3 HBO treatments, the cells were rinsed with PBS, treated with 0.05% trypsin, and collected by centrifugation at 800× *g*. NE-PER nuclear and cytoplasmic extraction reagents (Thermo, USA) were used to isolate cytoplasmic and nuclear extracts from the cells. The protein content was quantitated using a protein assay kit (Pierce, Appleton, WI, USA), and separated by 7.5% SDS-PAGE to detect β-catenin (Millipore, USA) and TATA binding protein (TBP; Abcam).

### 2.11. RNA Extraction and Real-Time PCR Detection of Wnt3a, MMP-3, and MMP-9

At 24 h after 3 HBO treatments, cellular RNA was isolated using an RNeasy Mini Kit (Qiagen, CA, USA) and reverse-transcribed into cDNA with the ImProm-II Reverse Transcription System (Promega). For real-time PCR detection of Wnt3a, MMP-3, and -9 RNA transcripts, cDNA was analyzed on an ABI PRISM 7900 Sequence Detection System using TaqMan PCR Master Mix (Applied Biosystems). The cycle threshold (Ct) values were obtained, and the data were normalized to GAPDH expression using the 2^−ΔΔCt^ method to calculate the relative mRNA levels of each target gene.

### 2.12. MMPs ELISA Assay

The levels of MMP-3 and -9 in condition media (CM) after hyperbaric or normobaric treatments were determined using a commercial immunoassay kit (Quantikine Human MMP-3 and MMP-9; R & D Systems). At 24 h after each treatment, 200 μL of CM was sampled and analyzed according to the manufacturer’s instructions. All measurements were performed in duplicate. The results were normalized to μg DNA. The DNA content was determined using DNAzol Reagent (DNAzol; Invitrogen, MD, USA) according to the manufacturer’s instructions.

### 2.13. Effect of HBO on Rabbit IVD Degeneration Model

#### 2.13.1. Rabbit IVD Degeneration Model

We used the external compression device that was developed by Kroeber et al. [36] to create IVD degeneration in a rabbit model. Briefly, six New Zealand rabbits (skeletally mature, 3.2 ± 0.4 kg) were anesthetized by injection of a 5 mL mixture (1:1) of Zoletil and Rompun. With the six rabbits under general anesthesia, through a dorsal approach to the lumbar spine, the custom-made external loading device was attached using four stainless steel pins inserted into the vertebra body L4 and L5 parallel to the adjacent study disc by a variable-speed electric drill. After the wound was closed, axial stress to the disc was immediately created by a calibrated spring within the loading device to produce a disc compressive force in the animals.

#### 2.13.2. Effect of HBO on Degenerated Rabbit IVD

Six IVD degenerated rabbits were used and divided into two groups: (I) HBO group, which included 3 rabbits exposed to HBO with 100% oxygen at 2.5 ATA for 1.5 h daily, 5 days a week after surgery in a HBO animal chamber (Perry Baromedical Corporation, Riviera Beach, FL, USA), and (II) control group, housed in cages containing normal air. After 4 weeks of mechanical loading, the custom-made external loading devices were removed and Safranin-O staining was used to identify disc degeneration.

#### 2.13.3. Antigenic KS Detection in Blood Serum

At intervals of 1, 2, and 4 weeks after operation, venous blood was obtained from the rabbit ears in the control and HBO groups. Blood serum was obtained after centrifugation and analyzed for agKS (antigenic KS, bearing the 1/20/5-D-4 epitope) content by competitive indirect ELISA [24,37]. Briefly, after an inhibition step in which the 1/20/5-D-4 monoclonal antibody was allowed to interact with agKS in a diluted sample of serum, the mixture was placed in the well of a microtiter plate coated with chondroitinase ABC treated, agKS-containing aggrecan. After rinsing, the mixture was incubated with a solution of peroxidase coupled with anti-mouse IgG. Then, after further rinses, the new mixture was incubated with a solution containing the substrate for peroxidase. The absorbance of the colored product was read at 490 nm. The concentration of agKS was calculated by comparing this absorbency value with that generated using serial dilutions of an international standard of purified bovine KS (Sigma, USA). The KS standard was used at concentrations ranging from 6.25 to 200 ng/mL.

#### 2.13.4. Morphological Observation and Immunohistochemically Examination for Wnt3a and β-Catenin

The degenerated IVD was sent for immunohistochemically analysis. Tissue blocks were fixed in 10% formalin, decalcified with 20% EDTA, and embedded in paraffin. Five-micron sections were cut and stained with Safranin-O. Tissue sections were deparaffinized and incubated with proteinase K (25 μg/mL, Sigma) for 60 min at 37 °C. Endogenous peroxidase activity was blocked with 1% hydrogen peroxide. The presence and distribution of Wnt3a and β-catenin was determined using antibodies to detect Wnt3a (1:200 dilution, Santa Cruz Biotechnology) and β-catenin (1:40 dilution, BD Transduction Laboratories) with exposure at 4 °C overnight. Subsequently, a HRP linking 2° Ab was used for 30 min. Bound immunoglobulin was detected using a Dako REAL Envision Detection System (Dako, Carpinteria, CA, USA) and 0.1% methyl green (Dako) was used for counterstaining.

### 2.14. Statistical Analysis

Each human disc yielded one sample. The control and HBO samples separated from the same (Control group: without HBO treatment) so we used a paired *t*-test to analyze the control/HBO ratio in this study. Data were represented as mean ± standard division (SD). The *p*-values for the paired Student’s *t*-test were performed using the SPSS software package (Version 12.0, Chicago, IL, USA, SPSS Inc.). A *p*-value of <0.05 was considered statistically significant.

## 3. Results

### 3.1. Heat Maps of MiR Expression in Degenerated NPCs following Treatments

There were 116 miRNAs upregulated and 67 downregulated by at least 1.5-fold following HBO intervention (Figure 1a, *n* = 5). MiR-107 was chosen for further investigation (Figure 1b) as previous studies revealed that miR-107 mediates an inflammation response [38] and its expression is lower in IDD tissue [39].

### 3.2. HBO Upregulated MiR-107 Expression in Degenerated NPCs

HBO upregulated miR-107 expression in degenerated NPCs (HBO/control, 6.44 ± 1.67-fold, ** *p* < 0.01, *n* = 5; Figure 1c). Our data suggested that miR-107 might play an important role in suppressing the progression of IDD following HBO treatment.

#### 3.2.1. Seed Sequence of miR-107 in the 3′ UTR of the Wnt3a mRNA

To find the possible molecular targets of miR-107, we screened for putative target genes of miR-107 using (a) Target Scan 7.2 and (b) miRnalyze software. We found that Wnt3a, an important stimulator of canonical pathway, was likely a direct target of miR-107, as the 3′ UTR of Wnt3a contained a potential binding element for miR-107 with a 7-nt match to the miR-107 seed region (Figure 2a,b). Furthermore, cross-species conservation of the miR-107 seed sequence in the 3′ UTR of the Wnt3a mRNA was confirmed by the Target Scan algorithm (Figure 2c). These results indicated that hsa-miR-107 might target the Wnt3a mRNA via directly recognizing its seed-matched sequence in the 3′ UTR.

#### 3.2.2. Wnt3a Is a Direct Target of miR-107

In order to validate the direct targeting of Wnt3a by miR-107, the wild type (WT) or a mutant variant (Mut) of the Wnt3a 3′ UTR containing the target sequence was cloned into a dual-luciferase reporter vector (Figure 3a). Overexpression of miR-107 following HBO treatment significantly inhibited luciferase activity of the WT Wnt3a 3′ UTR (HBO/control, 0.65 ± 0.09-fold, ** *p* < 0.01, *n* = 4), whereas mutation of the miR-107 binding sites abolished this inhibitory effect of miR-107 (1.01 ± 0.09-fold, *p* > 0.05, *n* = 4) in the degenerated NPCs (Figure 3b). These results support the conclusion that Wnt3a is a target gene of miR-107 following HBO intervention.

We next examined the expression of Wnt3a in NPCs transfected with an anti-miR-107 construct following HBO treatment. HBO downregulated the mRNA expression of Wnt3a (HBO/control, 0.37 ± 0.05-fold, *** *p* < 0.001, *n* = 4) whereas transfection with anti-miR-107 upregulated the mRNA level of Wnt3a in NPCs following HBO intervention (HBO/control, 0.72 ± 0.09-fold, * *p* < 0.05, *n* = 4; Figure 3c). Western blot analysis was performed to examine the protein level of Wnt3a (Figure 3d), and the results showed that HBO led to a significant decrease in the protein level of Wnt3a (HBO/control, 0.50 ± 0.03-fold, ** *p* < 0.01, *n* = 3), whereas knockdown of miR-107 partly reversed Wnt3a protein expression in NPCs following HBO intervention (HBO/control, 0.67 ± 0.06-fold, * *p* < 0.05, *n* = 3). These data indicate that Wnt3a was negatively mediated by miR-107 at the post-transcriptional level in NPCs following HBO treatment, as overexpression of miR-107 following HBO intervention significantly inhibited the mRNA (Figure 3c) and protein (Figure 3d) expression of Wnt3a in these cells.

### 3.3. Effects of HBO on Wnt3a, LRP6 Phosphorylation, β-Catenin Translocation, and Cyclin D1 Expression

The Wnt3a/β-catenin pathway is initiated through Wnt3a interaction with Frizzled and LRP5/6. LRP5/6 are single transmembrane–spanning cell surface receptors for Wnt-ligands. We determined Wnt3a/β-catenin signaling kinetics by detecting phosphorylated LRP6 and β-catenin upon HBO stimulation. The Western blot data showed that the protein levels of Wnt3a (HBO/control, 54.2% ± 11.1%, * *p* < 0.05, *n* = 3) and phosphorylated LRP6 (HBO/control, 35.6% ± 5.0%, ** *p* < 0.01, *n* = 3) were downregulated after culturing for 5 d with HBO intervention (Figure 4a). Additionally, the suppression of the Wnt3a pathway resulted in a downregulated expression of a Wnt3a target gene, the protein cyclin D1 (HBO/control, 59.5% ± 9.5%, * *p* < 0.05, *n* = 3; Figure 4a). Next, we examined the influence of HBO on β-catenin nuclear translocation. After the extraction from cytoplasm and nucleus, the protein levels of total β-catenin (HBO/control, 44.0% ± 3.6%, ** *p* < 0.01, *n* = 3; Figure 4a) and that of β-catenin in the nuclear fractions (HBO/control, 30.4% ± 6.8%, ** *p* < 0.01, *n* = 3; Figure 4b) were downregulated after HBO treatment. HBO decreased the translocation of β-catenin from the cytosol into the nucleus and demoted the expression of cyclin D1.

### 3.4. Effects of HBO on the Activity of the Wnt/β-Catenin Signaling

To evaluate the effects of HBO on the activity of the Wnt/β-catenin signaling, we analyzed the activity of both TOP flash (containing the wild-type TCF binding sites) and FOP flash (mutant TOP flash) in NPCs following HBO treatment. Figure 5 shows that there was decreased TOP flash activity following HBO stimulation (control: 1.05 ± 0.07-fold vs. HBO: 0.52 ± 0.09-fold, ** *p* < 0.01, *n* = 3), whereas the FOP flash activity (control: 1.03 ± 0.02-fold vs. HBO: 1.06 ± 0.06-fold, *p* > 0.05, *n* = 3) was not affected. HBO downregulated the activity of the Wnt/β-catenin signaling.

### 3.5. Effects of HBO on the mRNA Expression of MMPs

MMPs are transcriptional targets of the canonical Wnt pathway. At 24 h after three HBO treatments, Wnt-target MMPs such as MMP-3 (HBO/control, 45.0% ± 17.3%, *** *p* < 0.001, *n* = 4) and MMP-9 (HBO/control, 27.0% ± 9.1%, *** *p* < 0.001, *n* = 4, Figure 6a) were transcriptionally downregulated in the degenerated NPCs following HBO treatment.

### 3.6. Effects of HBO on MMPs Secretion

Figure 6b,c and Table 1 present the effects of HBO on MMP-3 and -9 protein secretion by degenerated NPCs. HBO significantly decreased the secretion of MMP-3 (Figure 6b and Table 1) and -9 (Figure 6c and Table 1) by NPC after two or three HBO interventions. HBO downregulated the Wnt signaling, contributing to the regulation of MMPs secretion.

### 3.7. Effects of HBO on Degenerated Rabbit IVD

External axial loading in the rabbit IVD was used to induce disc degeneration (Figure 7). After loading, disorganization of the architecture of the annulus occurred. Safranin-O staining showed that mechanical loading induced loss of proteoglycan (PG), calcification of the IVD, and loss of the boundary between the AF and NPCs (Figure 8a). HBO suppressed the PG loss induced by mechanical loading (Figure 8b) as compared with the control group (Figure 8a).

The serum keratan sulfate (KS) levels in both groups are shown in Figure 8c and Table 2. No significant differences were noted in pre-operative serum KS levels between the control and HBO groups (*p* > 0.05, *n* = 3). After mechanical loading, serum KS levels rose sharply in week 1, then dropped in week 2 and week 4 in the control group. Serum KS levels were also increased in the HBO group in week 1, but were significantly lower than that of the control group 2 (* *p* < 0.05, *n* = 3) and 4 weeks (* *p* < 0.05, *n* = 3) after loading. Serum KS level was significantly increased after mechanical loading and gradually returned to near baseline after HBO treatment (Figure 8c and Table 2).

### 3.8. Effects of HBO on Wnt3a and β-Catenin Expression in Degenerated IVD Tissue

In the repaired IVD tissue, a reduced staining intensity ratio of Wnt3a (as a percent of the positive stained area) was observed in the cytoplasm and cartilage matrix of the HBO group (Figure 9b, 4.5% ± 0.72%) as compared to the control group (Figure 9a, 26.2% ± 4.2%; *** *p* < 0.001, *n* = 3). In addition, stronger expression was observed in the β-catenin positive staining intensity ratio in the control group (Figure 9c, 3.5% ± 0.81%) as compared to the HBO group (Figure 9d, 1.0% ± 0.63%; * *p* < 0.05, *n* = 3). IHC demonstrated that HBO treatment markedly suppressed Wnt3a and β-catenin expression in repaired IVD tissue.

## 4. Discussion

MiR-107 regulates gene expression involved in cell division [40], hypoxic stress response [41,42], and angiogenesis [43] in different tissues and cells. The effect of oxygen on miR-107 expression is cell type dependent. MiR-107 expression was upregulated by hypoxia in human cancer cell lines [41]. However, miR-107 expression upregulation in the retina following hyperoxia treatment has also been reported [44]. Although IVD is a hypoxic tissue, the oxygen concentration showed a further decrease in degenerated IVD compared to healthy IVD [22,23] and MiR-107 is downregulated in degenerated IVD [22]. In order to identify the miRNAs involved in the molecular regulation of NPCs following HBO intervention, the miRNA expression profile of NPCs was performed. In this study, miR-107 was one of the identified miRNAs upregulated in human degenerated NPCs following HBO treatment via microarray analysis (Figure 1a,b) and this expression was confirmed by real-time PCR (Figure 1c). Because miR-107 is downregulated during hypoxia [41], HBO increases the oxygen levels in culture media to improve hypoxia conditions, and therefore upregulate the expression levels of miR-107 in degenerated NPCs.

MicroRNAs are highly specific regulators of genes at the post-transcriptional level. They effectuate a sort of “fine-tuning” of gene expression by reducing the latter or by silencing genes, either inhibiting translation or causing the degradation of RNA [45,46]. In the present study, we investigated the effect of HBO on miR-107, which was found downregulated in the degenerative IVD cells (Figure 1). The Wnt3a mRNA which contained a matching sequence for miR-107 were decreased (Figure 2). As in the following study, transfection with an antagonist of miR-107 partly suppressed the effects of HBO (Figure 3). Wnt3a was negatively mediated by miR-107 at the post-transcriptional level in NPCs following HBO treatment, as overexpression of miR-107 following HBO intervention significantly inhibited the mRNA (Figure 3c) and protein (Figure 3d) expression of Wnt3a in these cells.

Wnt is an acronym combining the Int1 gene found in mice and the wingless gene from drosophila [47,48]. There are 19 Wnt ligand proteins in mammals, which are secreted and can act in an autocrine or paracrine manner by binding to a multitude of receptors and co-receptors [49]. This offers a temporal and spatial control for a variety of functions, including determination of cell proliferation and migration, regulation of cell survival, and differentiation. The multifaceted nature of the system ensures full control of the development process. This is achieved not only through spatial control of Wnt ligand expression, but also through a flexible combination of receptors and co-receptors expressed on physiological and abnormal target cells. The network of interactions is not fully understood; however, much data already exists [50].

Wnt pathways are usually segregated into three divisions. The canonical pathway is the first and most intensely studied pathway involving the ligands Wnt1, Wnt2, Wnt3, and Wnt8. These ligands bind to LRP5/6 co-receptors along with receptors of the Frizzled (*FZD*) family, which consists of 10 members. Upon binding to FZD/LRP receptor complexes, components of the steady degradation complex for β-catenin—including Dvl, Axin, Gsk-3β, and others—are recruited to the receptors, and β-catenin degradation is blocked, causing its accumulation and translocation to the nucleus where it activates Tcf/Lef-dependent transcription of target genes [51].

The expression of Wnt3a, GSK-3β, cyclin D1, and β-catenin was notably augmented in parallel with IDD progression [9]. MiR-107 is downregulated in degenerated IVD [22] and Wnt3a was identified as a target gene of miR-107 in the present study (Figure 2 and Figure 3). These results suggest that miR-107 might regulate signaling via a mechanism targeting Wnt3a. Since Wnt3a has been shown to be expressed in human and rat IVD tissue sections [6] and rat NPCs have been shown to exhibit increased levels of β-catenin under the hypoxic condition [52], we first investigated the role of the Wnt3a ligand in human degenerated NPCs following HBO treatment. The Q-PCR and Western blot analyses revealed that Wnt3a was expressed in human degenerated NPCs, that HBO significantly decreased the mRNA (Figure 3c) and protein (Figure 3d) expressions of Wnt3a as compared with the control cells, and that miR-107 inhibitors reversed the suppressive effects of HBO. The canonical Wnt/β-catenin signaling pathway is implicated in an array of cellular pathological processes, including cell proliferation, apoptosis, and differentiation [53]. In particular, the involvement of Wnt/β-catenin signaling in the pathogenesis of IDD has been currently revealed. However, different researches have arrived at contradictory results considering the role of Wnt signaling in mediating this degenerative disease. Smolders et al. reported that the downregulated Wnt signaling was involved in the early stage of IDD development in the chondrodystrophic dog model [54]. Sun et al. found that Wnt/β-catenin was suppressed by miR-532 and that suppression of Wnt/β-catenin signaling remarkably boosted miR-532-induced NPC cell apoptosis [55]. Shh signaling and canonical Wnt signaling pathways are downregulated in adult IVDs and this downregulation is reversible [56]. Inversely, Hiyama et al. indicated that Wnt signaling activation retarded NPCs proliferation and promoted cell senescence, that Wnt/β-catenin signals contribute to the pathogenesis of IVD degeneration, and that activation of Wnt/β-catenin signaling may lead to an increased breakdown of the matrix and might cause the degeneration of NPCs [13,14,15]. Beyond these contradictory results, the important role of Wnt signaling pathway was shown in IDD development. In the present study we noted that Wnt/β-catenin signaling was downregulated by miR-107 following HBO treatment in NPCs (Figure 2 and Figure 3). This implies that HBO may exert the protective effects on NPCs through inhibition of Wnt3a/β-catenin pathway.

Wnt/β-catenin signaling triggers the process of IVD degeneration [13,14,15]. A key step after Wnt stimulation is to phosphorylate the LRP6 intracellular domain, and Wnt-induced LRP6 phosphorylation has been shown to initiate Wnt/β-catenin signaling [54,55]. Our Western blot data showed that the protein level of phosphorylated LRP6 was downregulated after culturing for 5 d with HBO treatment (Figure 4a). HBO treatment significantly decreased the expression of Wnt3a (Figure 3c,d), and thus decreased the Wnt-induced LRP6 phosphorylation in the degenerated NPCs (Figure 4a).

Oxygen reaches the NP predominantly through diffusion, thereby imposing a hypoxic state on the NP cells [18,19], which is in turn enhanced by the loss of CEP permeability during IVD degeneration [20,21]. Because rat NPCs were shown to exhibit increased levels of β-catenin under the hypoxic condition [52] and β-catenin protein was upregulated in IDD tissue [5], we used HBO to improve hypoxia conditions and found that HBO resulted in decreased β-catenin protein levels in the degenerated NPCs (Figure 4a). In addition, the protein levels of β-catenin in the nuclear fractions were downregulated after HBO treatment. HBO decreased the translocation of β-catenin from the cytosol into the nucleus (Figure 4b); β-catenin may interact with TCF/LEF transcription factors and regulated Wnt3a target genes [4,43] and demoted the expression of cyclin D1 (Figure 4a).

To further evaluate the activation of β-catenin signaling, we measured the basal activity of the TOP flash reporter in the NPCs. The TOP/FOP flash luciferase reporter assay has been developed as a reliable, substantially amplified readout of Wnt/β-catenin pathway activation. Figure 5 shows that there was decreased TOP flash activity following HBO stimulation, whereas the FOP flash activity was not affected. The activation of the TOP flash reporter is known to be specific to the Wnt3a genes [30,57]. HBO decreased the expression of Wnt3a, which suppressed β-catenin–TCF transcriptional activity in this study. NPCs from degenerated discs have been reported to produce catabolic and inflammatory factors, including IL-1β, NO, and MMPs [1,2]. We previously reported that HBO decreases the expression of MMP-3 in degenerated NPCs [58]. Because MMP-3 and -9 are transcriptional targets of the canonical Wnt pathway [8,38], we investigated the mechanisms controlling the Wnt3a/β-catenin signaling and the transcriptional activation in degenerated NPCs following HBO treatment. Our results suggested that Wnt-target MMPs such as MMP-3 and -9 were transcriptionally downregulated in the degenerated NPCs after HBO treatment (Figure 6a). HBO decreased Tcf-dependent transcription and suppressed the expression of Wnt-target MMPs such as MMP-3 and -9. In addition, the protein levels of MMP-3 and -9 (Figure 6b) secreted by degenerated NPCs were also downregulated following HBO treatment.

The effects of HBO on cells are dose dependent. In previous experiments, De Bels et al. showed hyperoxia incubation lasting for 18 h and the medium was kept unchanged. Because of long term incubation and without an air break, hyperoxia altered the ultrastructure and induced apoptosis in leukemia cell lines [59]. However, de Wolde et al. observed decreased neutrophilic ROS production and phagocytosis following the second HBO therapy session, which persisted after the third session, but no alterations in MDA concentrations [60]. In the present study, we had similar HBO protocol to de Wolde et al. All hyperoxic cells were exposed to three HBO sections (25 min × 3) with two 5 min breaks of breathing normal air per HBO treatment. A series of three HBO treatments (Day 1, Day 3, and Day 5) was used in a week. At 24 h after each treatment, conditioned media (CM) were collected for further analysis by ELISA. We replaced the culture medium in the dishes after the first and second CM collection. It is important to note, changing the medium will also change the oxygen content [59].

Matrix metallopeptidase 9 (MMP-9) is an enzyme that belongs to the zinc metalloproteinases family involved in the degradation of the extracellular matrix. MMPs can cleave or degrade clotting factors, chemotactic molecules, cell–cell adhesion molecules, latent growth factors, and cell surface receptors. Out of 24 MMPs, MMP-9 is the only one that is both undetectable in healthy tissues and highly expressed in inflammation and in several diseases, including cancer [61]. Pulsed hyperoxic treatment modulates MMP-9 release and activity in healthy human plasma [61]. The activity of MMP-9 was significantly downregulated during hyperoxia and rapidly reaching a level significantly higher than that monitored at the baseline after 4 h of normoxic treatment. The upregulation of the catalytic activity of MMP-9 after the return to normoxia supports the hypothesis that normoxia following a hyperoxic event is sensed by endothelial cells as an oxygen shortage [62]. In the present study, MMP-9 was more highly expressed in degenerated IVD cells than in healthy condition. Our data suggest that the degenerated IVD cells became healthier and downregulated MMP-9 expression following HBO treatment. Because de Wolde et al. observed decreased neutrophilic ROS production and phagocytosis following the second HBO therapy session, which persisted after the third session, but no alterations in MDA concentrations [60], we used a similar HBO protocol with three sections of HBO treatment.

Activation of Wnt/β-catenin has been reported to inhibit NPCs proliferation and trigger cell senescence, thereby promoting IDD aggravation [8]. Li et al. placed rat NPCs under a pressure of 1.0 MPa and found that the Wnt/β-catenin signaling pathway was activated in a time-dependent manner [63]. Mechanical loading induced IVD degeneration and tended to upregulate the markers of Wnt signaling and β-catenin expression in IVD. Additionally, mechanical loading also increased catabolic markers MMP-3 and -13 expression in IVD [63]. In the present study, we used the rabbit external compression model developed by Kroeber et al. [36] to investigate the ability of HBO to stimulate disc regeneration (Figure 7). Mechanical compression of the rabbit IVD induced severe PG degradation, calcification of the IVD, and loss of the demarcation between the NP and AF (Figure 8a). Excessive loading supports upregulation of matrix-degrading enzymes, rather than a direct stimulus for matrix degradation [64,65]. Mechanical compression increased catabolic markers of Wnt signaling in IVD [66]. HBO suppressed the Wnt3a/β-catenin signaling induced by mechanical loading (Figure 9b,d), thus suppressing PG loss in IVD (Figure 8b). Serum KS serves as a potential biomarker of loading of the IVD [35]. Serum KS level was significantly increased after mechanical loading and gradually returned to near baseline after HBO intervention (Figure 8c).

Our data suggest that HBO inhibiting the Wnt3a/β-catenin signaling pathway by upregulating miR-107 expression in human degenerated disc may be feasible as a therapeutic for degenerative disc disease. However, there are some restrictions in this study. Because other miRNAs may also be relevant for the regulation of Wnt signaling in NPCs following HBO intervention, transfection with anti-miR-107 only partly suppressed the effects of HBO treatment (Figure 3). In the future, further systematic and in-depth experiments investigating the Wnt signaling following HBO treatment will be conducted. The positive effects of HBO on various conditions have been studied both in animal experiments and in clinical use [67,68]. Recently, HBO therapy in clinical patients with full endoscopic lumbar diskectomy (FELD) has been reported [69]. HBO may provide a novel therapeutic strategy for IDD. The clinical applications of HBO in degenerative IVD diseases need to be further investigated.

## 5. Conclusions

HBO inhibits Wnt3a/β-catenin signaling-related pathways by upregulating miR-107 expression in degenerated NPCs. HBO may play a protective role against IVD degeneration and could be used as a future therapeutic treatment.

## Figures and Tables

**Figure 1 cells-12-02111-f001:**
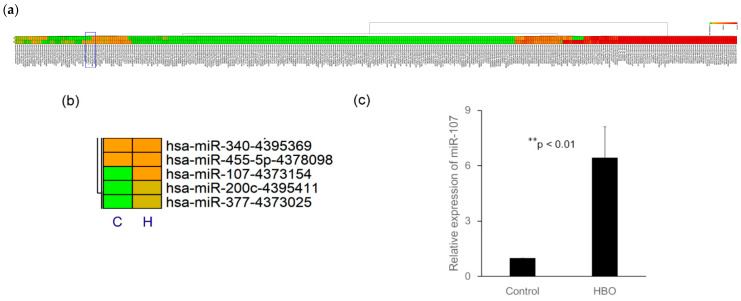
**HBO modulates the expression of miRNAs in degenerated NPCs.** (**a**) A heat map of the miRNAs with significantly changed expression levels in the HBO-treated group compared with the control group. Average expression of each miRNA in each class is shown. There were 116 miRNAs upregulated and 67 downregulated by at least 1.5-fold following HBO treatment. (**b**) MiR-107 was chosen for further investigation. C: control, H: HBO. (**c**) HBO treatment increased miR-107 expression in degenerated NPCs (** *p* < 0.01 vs. Control, *n* = 5).

**Figure 2 cells-12-02111-f002:**
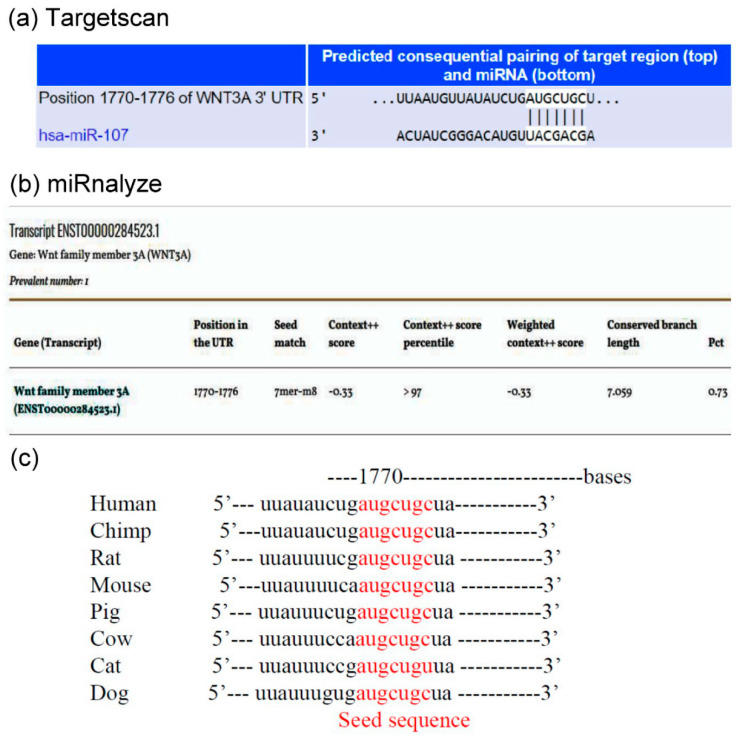
**Seed sequence of miR-107 in the 3′ UTR of the Wnt3a mRNA.** (**a**) TargetScan predicted the duplex of miR-107 with the seed sequence in the 3′ UTR of the human Wnt3a mRNA. The sequences in white are the locations of the potential seed-matched sequences for the miRNAs assessed. (**b**) MiRnalyze predicted the duplex of miR-107 with its seed sequence in the 3′ UTR of the human Wnt3a mRNA. (**c**) Cross-species conservation of the miR-107 seed sequence in the 3′ UTR of the human Wnt3a mRNA as identified via the TargetScan algorithm (sequences in red).

**Figure 3 cells-12-02111-f003:**
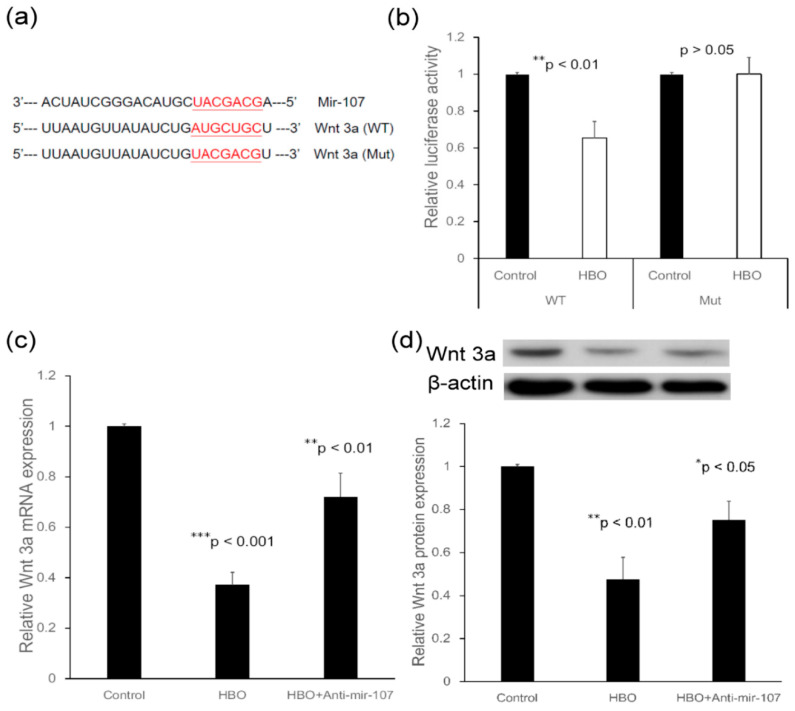
**Wnt3a is a direct target of miR-107.** (**a**) Diagram of the binding site between miR-107 and the Wnt3a 3′-UTR. The reporter vectors contain the wild type (WT) or mutant of Wnt3a 3′-UTR. (**b**) Dual-luciferase reporter assay of Wnt3a 3′-UTR. The reporter vectors containing the WT or mutant of Wnt3a 3′-UTR were transfected into NPCs. Luciferase activity was significantly downregulated after HBO treatment (** *p* < 0.01 vs. control, *n* = 4) in the WT but not in the mutant type (*p* > 0.05 vs. control, *n* = 4). (**c**) Real-time PCR analysis of Wnt3a mRNA expression in NPCs transfected with miR-107 inhibitors. Wnt3a mRNA expression was downregulated after HBO treatment (*** *p* < 0.001 vs. control, *n* = 4). MiR-107 inhibitors partly reversed the suppressive effects of HBO (** *p* < 0.01 vs. control, *n* = 4). (**d**) Western blot analysis of Wnt3a protein expression in NPCs transfected with miR-107 inhibitors. Values were normalized against β-actin. Wnt3a protein expression was significantly downregulated after HBO treatment (** *p* < 0.01 vs. control, *n* = 3). MiR-107 inhibitors partly reversed the suppressive effects of HBO (* *p* < 0.05 vs. control, *n* = 3).

**Figure 4 cells-12-02111-f004:**
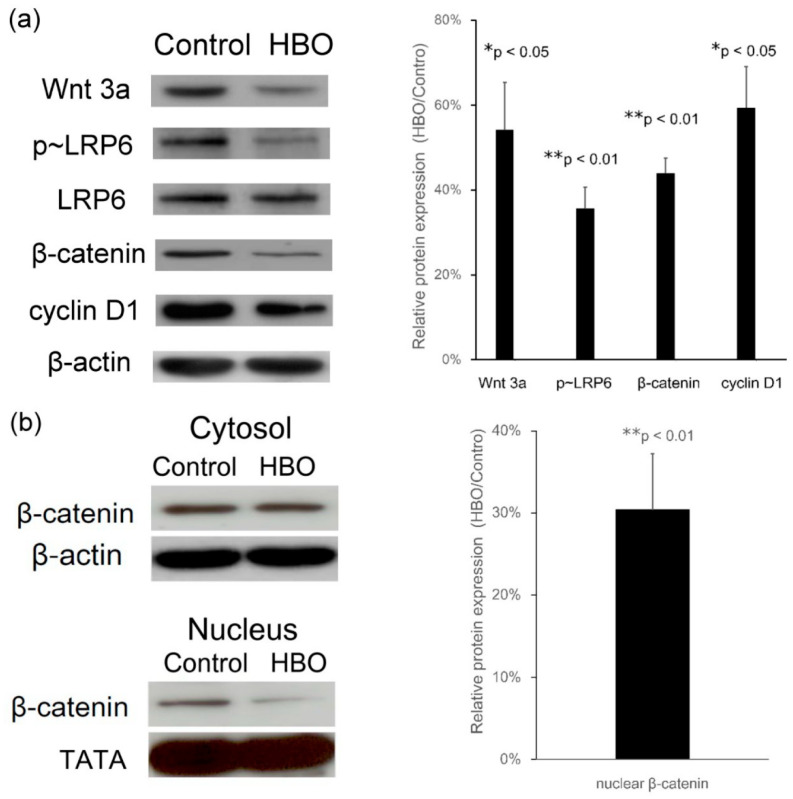
**Effects of HBO on LRP6 phosphorylation, β-catenin translocation**, **and cyclin D1 expression.** (**a**) HBO significantly decreased Wnt3a protein (* *p* < 0.05), phosphorylated LRP6 (** *p* < 0.01), and β-catenin protein (** *p* < 0.01) expressions as compared with the control cells. In addition, the suppression of the Wnt3a pathway by HBO resulted in a downregulated expression of a Wnt3a target gene, the protein cyclin D1 (* *p* < 0.05). (**b**) The protein levels of β-catenin in the nuclear fractions were downregulated after HBO treatment (** *p* < 0.01). HBO decreased the translocation of β-catenin from the cytosol into the nucleus and demoted the expression of cyclin D1.

**Figure 5 cells-12-02111-f005:**
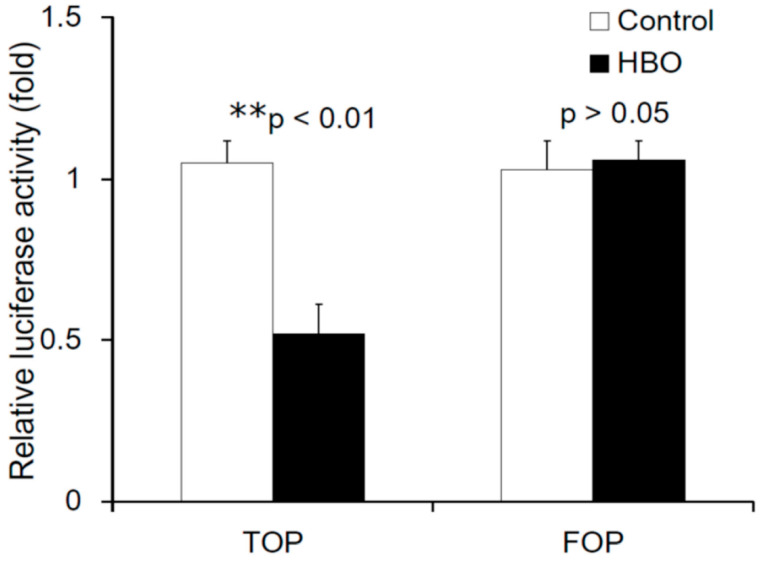
**Effects of HBO on the activity of the Wnt/β-catenin signaling.** TOP/FOP promoter assay was tested to determine Wnt signaling pathway activity. There was decreased TOP flash activity following HBO stimulation (** *p* < 0.01), whereas the FOP flash activity was not affected (*p* > 0.05). HBO treatment decreased the expression of Wnt3a, which suppressed β-catenin–TCF transcriptional activity.

**Figure 6 cells-12-02111-f006:**
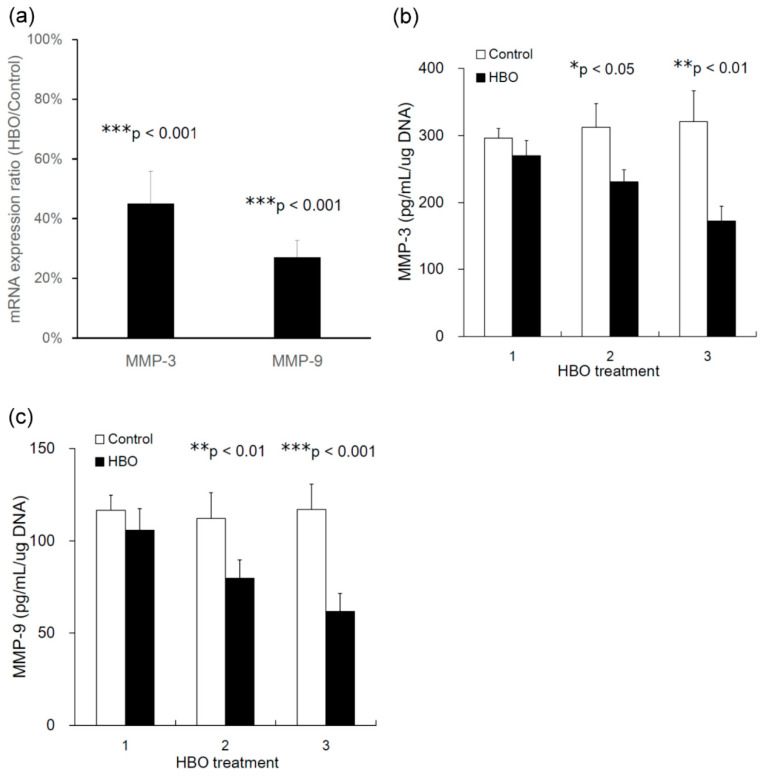
**Effects of HBO on mRNA expression and protein secretion of MMPs.** (**a**) After three HBO treatments, mRNA expression of Wnt-target MMPs such as MMP-3 (HBO/Control ratio, *** *p* < 0.001, *n* = 4) and MMP-9 (HBO/Control ratio, *** *p* < 0.001, *n* = 4) were transcriptionally downregulated in the degenerated NPCs following HBO treatment. (**b**) HBO significantly decreased the secretion of MMP-3 following two (* *p* < 0.05, *n* = 4) and three (** *p* < 0.01, *n* = 4) HBO treatments. (**c**) HBO significantly decreased the secretion of MMP-9 following two (** *p* < 0.01, *n* = 4) and three (*** *p* < 0.001, *n* = 4) HBO treatments.

**Figure 7 cells-12-02111-f007:**
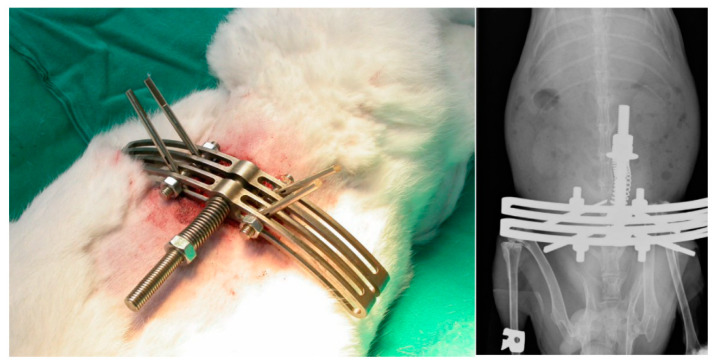
**External axial loading in the rabbit IVD.** An external compression device was attached to the four stainless steel pins inserted into the vertebra body L4 and L5 parallel to the adjacent study disc. External axial loading in the rabbit IVD was used to induce disc degeneration.

**Figure 8 cells-12-02111-f008:**
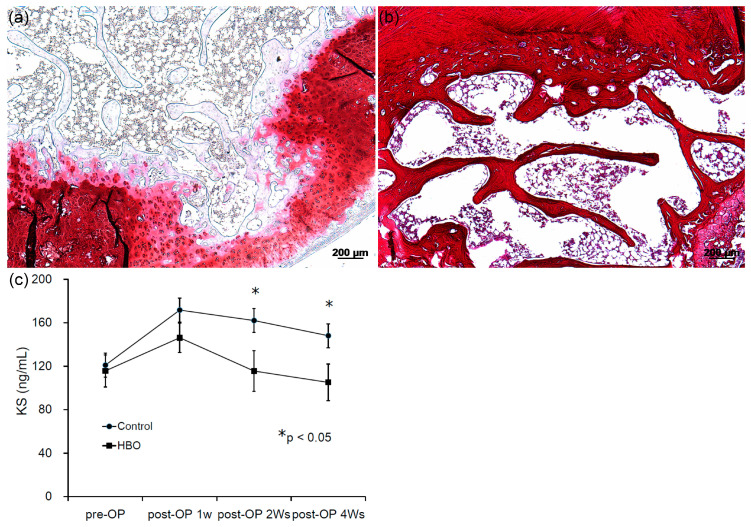
**Effects of HBO on degenerated rabbit IVD.** (**a**) Safranin-O staining showed that mechanical loading induced severe proteoglycan (PG) loss, calcification of the IVD, and loss of the demarcation between the NP and AF (×50 magnification). (**b**) HBO suppressed the PG loss induced by mechanical loading as compare with control group (×50 magnification). (**c**) No significant differences were noted in pre-operative serum KS levels between the control and HBO groups. After loading, serum KS levels rose sharply in week 1, and dropped in week 2 and week 4 in the control group. Serum KS levels were also increased in the HBO group in week 1, but were significantly lower than that of the control group 2 and 4 weeks after loading (* *p* < 0.05, *n* = 3).

**Figure 9 cells-12-02111-f009:**
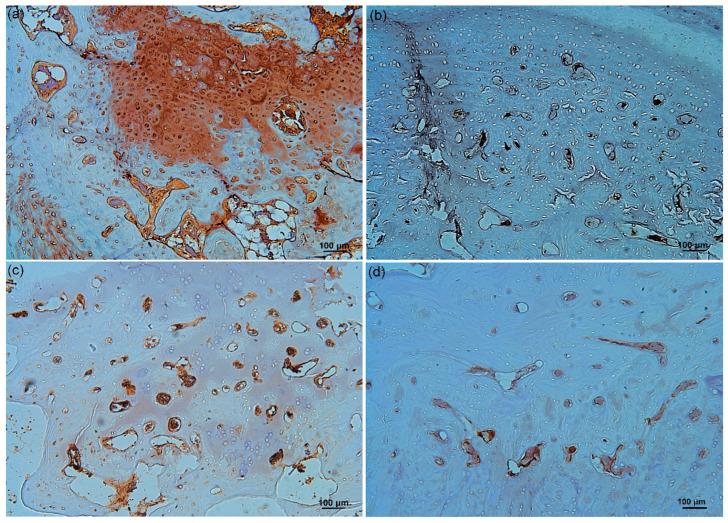
**Effects of HBO on Wnt3a and β-catenin expression in repaired IVD tissue.** In the repaired IVD tissue, a reduced staining intensity ratio of Wnt3a was observed in the cytoplasm and cartilage matrix of the HBO group (**b**) as compared to the control group (**a**). In addition, stronger expression was observed in the β-catenin positive staining intensity ratio in the control group (**c**) as compared to the HBO group (**d**). IHC demonstrated that HBO markedly suppressed Wnt3a and β-catenin expression in repaired IVD tissue (×100 magnification).

**Table 1 cells-12-02111-t001:** Effect of HBO on the extracellular release of MMPs by degenerated NPCs.

MMP-3	Control	HBO	*p*-Value
1st treatment	283.4 ± 28.2	260.5 ± 26.4	*p* > 0.05
2nd treatment	299.3 ± 38.8	227.3 ± 16.6	* *p* < 0.05
3rd treatment	316.3 ± 38.7	173.9 ± 19.1	*** *p* < 0.001
MMP-9	Control	HBO	*p*-value
1st treatment	115.5 ± 7.1	103.6 ± 10.6	*p* > 0.05
2nd treatment	113.9 ± 12.0	79.0 ± 8.3	** *p* < 0.01
3rd treatment	121.7 ± 14.7	61.3 ± 8.9	*** *p* < 0.001

Differences between control and HBO groups were analyzed using Student’s *t*-test. A *p*-value < 0.05 was considered statistically significant. * *p* < 0.05, ** *p* < 0.01, *** *p* < 0.001.

**Table 2 cells-12-02111-t002:** Effect of HBO on the rabbit’s serum KS level.

KS	Control	HBO	*p*-Value
pre-OP	121.0 ± 17.0	115.7 ± 14.7	*p* > 0.05
post-OP 1 Ws	171.7 ± 13.2	146.3 ± 13.6	*p* > 0.05
post-OP 2 Ws	162.1 ± 8.1	115.6 ± 18.8	* *p* < 0.05
post-OP 4 Ws	148.1 ± 9.4	105.1 ± 16.9	* *p* < 0.05

Differences between control and HBO groups were analyzed using Student’s *t*-test. A *p*-value < 0.05 was considered statistically significant. * *p* < 0.05.

## Data Availability

All data supporting our findings are contained within the manuscript.

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
