# Peer review of "Effects of Hyperbaric Oxygen Intervention on the Degenerated Intervertebral Disc: From Molecular Mechanisms to Animal Models"

_cells, 2023, doi:10.3390/cells12162111_

Round 1

Reviewer 1 Report

The authors are tackling a very interesting and largely uninvestigated phenomenon which is paramount in our society.

They went very deep in their analysis, opening useful and potentially groundbreaking new therapeutic applications.

I have some remarks and suggestions:

Abstract :         line 35-> try to replace the word consulted….-> observed..or any other synonym

Introduction : Line 54 -> I probably missed that but I think that you didn’t explain the Wnt acronym, as you did for Fz-LRP5/6 in the same sentence, I understand that Wnt is a known abbreviation, so it is for Fz-LRP5/6, I suggest to explain it somewhere, to suit readers that are not so acquainted with the field.

                        Line 101 Replace “author” with -> authors

Material and methods: When you harvest disks from patients, you may (due to surgery) elicit some reactions, due to the surgery and not only due to the degenerative status alone. You may add some explanations either in the methods or in the discussion about the mitigation of this putative bias ….for instance time of expression, care taken not to act on the degenerative part during surgery, meticulous excision before processing for analysis etc…etc… 

HBO -> When you use the HBO session, have you an idea of the oxygen content in de medium ? Did you change medium ? I think that it should be specified since changing medium will also change the oxygen content of the cells [1].

Results/Discussion

Very interesting data on luciferase and the discussion about Wnt. There are some ideas and newly published data showing the more that 3 sessions are needed to change some specific reaction (nevertheless, there are a lot of different reports also contradicting this) , it woul be of interest to explain the reader why 3 sessions have been chosen (may be because that was the time needed allowing not changing the medium….)[2,3]

I’m aware of some other references on hyperbaric oxygen MMP9 and Mirn expression as well as low oxygen pressure and cellular reactions that may be of interest [4-6]

1.         De Bels, D.; Tillmans, F.; Corazza, F.; Bizzari, M.; Germonpre, P.; Radermacher, P.; Orman, K.G.; Balestra, C. Hyperoxia Alters Ultrastructure and Induces Apoptosis in Leukemia Cell Lines. Biomolecules 202010, 282, doi:10.3390/biom10020282.

2.         de Wolde, S.D.; Hulskes, R.H.; de Jonge, S.W.; Hollmann, M.W.; van Hulst, R.A.; Weenink, R.P.; Kox, M. The Effect of Hyperbaric Oxygen Therapy on Markers of Oxidative Stress and the Immune Response in Healthy Volunteers. Front Physiol 202213, 826163, doi:10.3389/fphys.2022.826163.

3.         De Wolde, S.D.; Hulskes, R.H.; Weenink, R.P.; Hollmann, M.W.; Van Hulst, R.A. The Effects of Hyperbaric Oxygenation on Oxidative Stress, Inflammation and Angiogenesis. Biomolecules 202111, doi:10.3390/biom11081210.

4.         MacLaughlin, K.J.; Barton, G.P.; Braun, R.K.; MacLaughlin, J.E.; Lamers, J.J.; Marcou, M.D.; Eldridge, M.W. Hyperbaric air mobilizes stem cells in humans; a new perspective on the hormetic dose curve. Frontiers in Neurology 202314, doi:10.3389/fneur.2023.1192793.

5.         Fratantonio, D.; Virgili, F.; Zucchi, A.; Lambrechts, K.; Latronico, T.; Lafere, P.; Germonpre, P.; Balestra, C. Increasing Oxygen Partial Pressures Induce a Distinct Transcriptional Response in Human PBMC: A Pilot Study on the "Normobaric Oxygen Paradox". Int J Mol Sci 202122, 458, doi:10.3390/ijms22010458.

6.         Cimino, F.; Balestra, C.; Germonpre, P.; De Bels, D.; Tillmans, F.; Saija, A.; Speciale, A.; Virgili, F. Pulsed high oxygen induces a hypoxic-like response in human umbilical endothelial cells and in humans. J Appl Physiol (1985) 2012113, 1684-1689, doi:10.1152/japplphysiol.00922.2012.

 I hope that my remarks or suggestion will be helpful to the authors. Thank-you for giving me the opportunity to review such an interesting manuscript.

some minor changes

Author Response

Comments and Suggestions for Authors

I have some remarks and suggestions:

Abstract :     line 35-> try to replace the word consulted….-> observed..or any                        other synonym

Ans : I have replaced the word consulted to observed in Abstract.

Introduction : Line 54 -> I probably missed that but I think that you didn’t explain the Wnt acronym, as you did for Fz-LRP5/6 in the same sentence, I understand that Wnt is a known abbreviation, so it is for Fz-LRP5/6, I suggest to explain it somewhere, to suit readers that are not so acquainted with the field.

Ans:  Wnt is an acronym combining the Int1 gene found in mice and the wingless gene from drosophila [1,2]. There are 19 Wnt ligand proteins in mammals, which are secreted and can act in an autocrine or paracrine manner by binding to a multitude of receptors and co-receptors ([3]). This offers a temporal and spatial control for a variety of functions including determination of cell proliferation and migration, regulation of cell survival and differentiation. The multifaceted nature of the system ensures full control of the development process. This is achieved not only through spatial control of Wnt ligand expression, but also through a flexible combination of receptors and co-receptors expressed on physiological and abnormal target cells. The network of interactions is not fully understood, however, a lot of data already exists [4].

      Wnt pathways are usually segregated into three divisions. Canonical pathway is the first and most intensely studied pathway involves the ligands Wnt1, Wnt2, Wnt3, and Wnt8. These ligands bind to the LRP5/6 co-receptors together with receptors of the Frizzled (FZD) family, which consists of 10 members. Upon binding to FZD/LRP receptor complexes, components of the steady degradation complex for β-catenin—including Dvl, Axin, Gsk-3β and others—are recruited to the receptors, β-catenin degradation is blocked, causing its accumulation and translocation to the nucleus where it activates Tcf/Lef-dependent transcription of target genes [5].

Ref.

  1. Baker, N.E. Molecular cloning of sequences from wingless, a segment polarity gene in Drosophila: The spatial distribution of a transcript in embryos. EMBO J. 1987, 6, 1765–1773.
  2. Nusse, R.; Varmus, H. Three decades of Wnts: A personal perspective on how a scientific field developed. EMBO J. 2012, 31, 2670–2684.
  3. Niehrs, C. The complex world of WNT receptor signalling. Nat. Rev. Mol. Cell Biol. 2012, 13, 767–779.
  4. MacDonald, B.T.; Tamai, K.; He, X. Wnt/β-Catenin Signaling: Components,Mechanisms, and Diseases. Dev. Cell 2009, 17, 9–26.
  5. Clevers, H.; Nusse, R. Wnt/β-Catenin Signaling and Disease. Cell 2012, 149, 1192–1205.

The above description has been shown in Discussion.

Line 101 Replace “author” with -> authors

Ans : It has been replaced

Material and methods: When you harvest disks from patients, you may (due to surgery) elicit some reactions, due to the surgery and not only due to the degenerative status alone. You may add some explanations either in the methods or in the discussion about the mitigation of this putative bias ….for instance time of expression, care taken not to act on the degenerative part during surgery, meticulous excision before processing for analysis etc…etc… 

Ans: The indications of transforaminal lumbar interbody fusion (TLIF) by cage in our clinical practice were grade 1 spondylolisthesis with instability (change of Cobb’s angle > 11°, or change of slippage > 4mm in flexion and extension lateral lumbar spine radiograms; grade 2 or more spondylolisthesis; or segmental instability, all had certain degree of disc degeneration and were proofed to be Pfirmann grading III or more in preoperative MRI study. The studied disc material was harvested from the target disc of TLIF performed, and the soft disc tissue from the disc space instead of firm annulus part or hard osteochondral plate from endplates was collected.

The above description has been shown in Material and methods.

HBO -> When you use the HBO session, have you an idea of the oxygen content in de medium? Did you change medium ? I think that it should be specified since changing medium will also change the oxygen content of the cells [1].

Ans: The dissolved oxygen levels in culture medium maintained in standard culture conditions (about 18% O2, 1ATA, Control group). According to the Henry’s Law, the amount of gas dissolved in a liquid is directly proportional to the partial pressure of that gas. In our study, the amount of oxygen will increase in culture medium during HBO treatment (100% O2, 2.5ATA). Because DMEM / F12 containing phenol red indicator, the color of medium changed from red to pink after HBO treatment. When the pressure returned to normal condition, the dissolved oxygen level in culture medium returned to standard culture conditions (18% O2, 1ATA). A three HBO sections (25 minutes × 3) per HBO treatment. A series of 3 times of HBO treatment was used in a week. At 24 hours after each treatment, conditioned media (CM) were collected, remove debris, and stored at –70°C until further analysis. We replaced new culture medium to dishes after first and second CM collection. Actually, changing medium will also change the oxygen content [1].

Results/Discussion

Very interesting data on luciferase and the discussion about Wnt. There are some ideas and newly published data showing the more that 3 sessions are needed to change some specific reaction (nevertheless, there are a lot of different reports also contradicting this), it would be of interest to explain the reader why 3 sessions have been chosen (may be because that was the time needed allowing not changing the medium….)[2,3]

I’m aware of some other references on hyperbaric oxygen MMP9 and Mirn expression as well as low oxygen pressure and cellular reactions that may be of interest [4-6]

  1. De Bels, D.; Tillmans, F.; Corazza, F.; Bizzari, M.; Germonpre, P.; Radermacher, P.; Orman, K.G.; Balestra, C. Hyperoxia Alters Ultrastructure and Induces Apoptosis in Leukemia Cell Lines. Biomolecules 202010, 282, doi:10.3390/biom10020282.
  2. de Wolde, S.D.; Hulskes, R.H.; de Jonge, S.W.; Hollmann, M.W.; van Hulst, R.A.; Weenink, R.P.; Kox, M. The Effect of Hyperbaric Oxygen Therapy on Markers of Oxidative Stress and the Immune Response in Healthy Volunteers. Front Physiol 202213, 826163, doi:10.3389/fphys.2022.826163.
  3. De Wolde, S.D.; Hulskes, R.H.; Weenink, R.P.; Hollmann, M.W.; Van Hulst, R.A. The Effects of Hyperbaric Oxygenation on Oxidative Stress, Inflammation and Angiogenesis. Biomolecules 202111, doi:10.3390/biom11081210.
  4. MacLaughlin, K.J.; Barton, G.P.; Braun, R.K.; MacLaughlin, J.E.; Lamers, J.J.; Marcou, M.D.; Eldridge, M.W. Hyperbaric air mobilizes stem cells in humans; a new perspective on the hormetic dose curve. Frontiers in Neurology 202314, doi:10.3389/fneur.2023.1192793.
  5. Fratantonio, D.; Virgili, F.; Zucchi, A.; Lambrechts, K.; Latronico, T.; Lafere, P.; Germonpre, P.; Balestra, C. Increasing Oxygen Partial Pressures Induce a Distinct Transcriptional Response in Human PBMC: A Pilot Study on the "Normobaric Oxygen Paradox". Int J Mol Sci 202122, 458, doi:10.3390/ijms22010458.
  6. Cimino, F.; Balestra, C.; Germonpre, P.; De Bels, D.; Tillmans, F.; Saija, A.; Speciale, A.; Virgili, F. Pulsed high oxygen induces a hypoxic-like response in human umbilical endothelial cells and in humans. J Appl Physiol (1985) 2012113, 1684-1689, doi:10.1152/japplphysiol.00922.2012.

 Ans: The effects of HBO on cells are dose dependent. In previous experiments, De Bels D, et al showed hyperoxia incubation was last for 18 h and the medium was kept unchanged. Because of long term incubation and without air break, hyperoxia altered ultrastructure and induced apoptosis in leukemia cell lines [1]. However, de Wolde et al. observed that decreased neutrophilic ROS production and phagocytosis following the second HBO therapy session, which persisted after the third session, but no alterations in MDA concentrations [2]. In the present study, we had similar HBO protocol with de Wolde et al. All hyperoxic cells were exposed to a three HBO sections (25 minutes × 3) with two 5-min breaks of breathing normal air per HBO treatment. A series of 3 times of HBO treatment (Day1, Day3, and Day5) was used in a week. At 24 hours after each treatment, conditioned media (CM) were collected for further analysis by ELISA. We replaced new culture medium to dishes after first and second CM collection. Actually, changing medium will also change the oxygen content [1].

  Matrix metallopeptidase 9 (MMP-9), is an enzyme that belongs to the zinc metalloproteinases family involved in the degradation of the extracellular matrix. MMPs can cleave or degrade clotting factors, chemotactic molecules, cell-cell adhesion molecules, latent growth factors, and cell surface receptors. Out of 24 MMPs, MMP9 is the only one that is undetectable in healthy tissues and highly expressed in inflammation and in several diseases, including cancer [5]. Pulsed hyperoxic treatment modulates MMP-9 release and activity in healthy human plasma [5]. The activity of MMP-9 was significantly downregulated during hyperoxia and rapidly reaching a level significantly higher than that monitored at the baseline after 4 h of normoxic treatment. The upregulation of the catalytic activity of MMP-9 after the return to normoxia supports the hypothesis that normoxia following a hyperoxic event is sensed by endothelial cells as an oxygen shortage [6]. In the present study, MMP9 is highly expressed in degenerated IVD cells than in healthy condition. Our data suggested that the degenerated IVD cells were become more healthy and down-regulated MMP9 expression following HBO treatment. Because de Wolde et al. observed that decreased neutrophilic ROS production and phagocytosis following the second HBO therapy session, which persisted after the third session, but no alterations in MDA concentrations [2]. We had similar HBO protocol with de Wolde et al by 3 sections of HBO treatment.

The above description has been shown in Discussion.

Reviewer 2 Report

This paper investigates the effects of hyperbaric oxygen therapy on intervertebral disc degeneration in vitro and in vivo, focusing on miR-107/Wnt3a-βcatenin signaling. The results indicate that hyperbaric oxygen therapy may suppress Wnt3a/β-catenin signaling by upregulating intervertebral disc miR-107 expression, thereby inhibiting disc degeneration. We believe that this paper is one of the studies that provide evidence for the usefulness of hyperbaric oxygen therapy for intervertebral disc degeneration.

Although we believe that this is a very useful paper with a lot of detailed research, the following points need to be addressed.

The quality of the histological images in Figure 8 and Figure 9 is poor. Scale bars should be included. Especially in Figure 8a and 8b, the histological images should be replaced with ones that clearly show the structure of the annulus fibrosus, nucleus pulposus, and end plate. And it is unlikely that both are Safranin O stained under the same conditions. In addition, histological grading of disc degeneration should be performed to compare the differences between the HBO-treated group and the Control group.

The mechanism by which HBO treatment upregulates miR-107 should be mentioned in the Discussion with citations added.

Author Response

Although we believe that this is a very useful paper with a lot of detailed research, the following points need to be addressed.

The quality of the histological images in Figure 8 and Figure 9 is poor. Scale bars should be included. Especially in Figure 8a and 8b, the histological images should be replaced with ones that clearly show the structure of the annulus fibrosus, nucleus pulposus, and end plate. And it is unlikely that both are Safranin O stained under the same conditions. In addition, histological grading of disc degeneration should be performed to compare the differences between the HBO-treated group and the Control group.

 Ans. The histological images for Figure 8 and Figure 9 have been replaced. In addition, scale bars were shown in the new figures. Both are Safranin O stained under the same conditions. However, we did not find the proper histological grading scheme for classification of intervertebral disc degeneration for the present study.

  1. The mechanism by which HBO treatment upregulates miR-107 should be mentioned in the Discussion with citations added.

Ans. MicroRNA is highly specific regulators of genes at the post-transcriptional level. They effectuate a sort of “fine-tuning” of gene expression by reducing the latter or by silencing genes, either inhibiting translation or causing the degradation of RNA [1,2]. In the present study, we investigated the effect of HBO on miR-107 which was found downregulated in the degenerative IVD cells (Figure 1). The Wnt 3a mRNA of which contained a matching sequence for miR-107 were decreased (Figure 2). As in the following study, transfection with an antagonist of miR-107 partly suppressed the effects of HBO (Fig.3). Wnt3a was negatively mediated by miR-107 at the post-transcriptional level in NPCs following HBO treatment, as overexpression of miR-107 following HBO intervention significantly inhibited the mRNA (Figure 3c) and protein (Figure 3d) expression of Wnt3a in these cells.

Ref.

  1. Mohr, A.M.; Mott, J.L. Overview of microRNA biology.  Liver Dis.201535, 3–11.
  2. Lindenmann, J.; Kamolz, L.; Graier, W.; Smolle, J.; Smolle-Juettner, F.-M. Hyperbaric Oxygen Therapy and Tissue Regeneration: A Literature Survey. Biomedicines 2022, 10, 3145

Round 2

Reviewer 2 Report

This revised manuscript answers the reviewer's questions and addresses their concerns. It deserves attention and is worthy of publication in this journal.